# DUBing Primary Tumors of the Central Nervous System: Regulatory Roles of Deubiquitinases

**DOI:** 10.3390/biom13101503

**Published:** 2023-10-10

**Authors:** Thomas Klonisch, Susan E. Logue, Sabine Hombach-Klonisch, Jerry Vriend

**Affiliations:** 1Department of Human Anatomy and Cell Science, Rady Faculty of Health Sciences, Max Rady College of Medicine, University of Manitoba, Winnipeg, MB R3E 0J9, Canada; 2Department of Pathology, Rady Faculty of Health Sciences, Max Rady College of Medicine, University of Manitoba, Winnipeg, MB R3E 0J9, Canada; 3Department of Medical Microbiology & Infectious Diseases, Rady Faculty of Health Sciences, Max Rady College of Medicine, University of Manitoba, Winnipeg, MB R3E 0J9, Canada; 4CancerCare Research Institute, CancerCare Manitoba, Winnipeg, MB R3E 0J9, Canada

**Keywords:** brain tumor, glioma, neuronal system tumor, deubiquitinase (DUB), endoplasmic reticulum associated degradation (ERAD), immune response, therapeutic target, DNA repair

## Abstract

The ubiquitin proteasome system (UPS) utilizes an orchestrated enzymatic cascade of E1, E2, and E3 ligases to add single or multiple ubiquitin-like molecules as post-translational modification (PTM) to proteins. Ubiquitination can alter protein functions and/or mark ubiquitinated proteins for proteasomal degradation but deubiquitinases (DUBs) can reverse protein ubiquitination. While the importance of DUBs as regulatory factors in the UPS is undisputed, many questions remain on DUB selectivity for protein targeting, their mechanism of action, and the impact of DUBs on the regulation of diverse biological processes. Furthermore, little is known about the expression and role of DUBs in tumors of the human central nervous system (CNS). In this comprehensive review, we have used publicly available transcriptional datasets to determine the gene expression profiles of 99 deubiquitinases (DUBs) from five major DUB families in seven primary pediatric and adult CNS tumor entities. Our analysis identified selected DUBs as potential new functional players and biomarkers with prognostic value in specific subtypes of primary CNS tumors. Collectively, our analysis highlights an emerging role for DUBs in regulating CNS tumor cell biology and offers a rationale for future therapeutic targeting of DUBs in CNS tumors.

## 1. Introduction

The ubiquitin proteasome system (UPS) is a highly regulated and dynamic process that utilizes a three-step enzymatic cascade to attach small molecules of the ubiquitin family onto proteins to alter their function and/or mark ubiquitinated proteins for proteasomal degradation. The extensive Ubiquitin and Ubiquitin-like Conjugation Database (UUCD) lists enzymes involved in ubiquitin post-translational modification of proteins [1]. In eukaryotes, this includes 1 human ubiquitin-activating (E1) enzyme (although the literature recognized at least two E1 enzymes, UBA1 and UBA6), 43 E2 ubiquitin-conjugating enzymes, 468 enzymes with E3 ligase activity (further classified as those with RING, HECT, or UBR domains), 538 E3 ligase adaptors, and approx. 100 deubiquitinase enzymes (DUBs) [2,3]. Ubiquitin is one of several ubiquitin-like protein modifiers that also includes ubiquilins, SUMO, NEDD8, and ISG15 [4]. While cellular regulators in their own right, these post-translational modifiers can cross-communicate with ubiquitin through modifications or are being modified by (poly)ubiquitin [5]. Ubiquitin has eight ubiquitination sites, including seven lysine (K) residues (K6, K11, K27, K29, K33, K48, K63) and a primary amine at the N-terminus [5]. While mono-ubiquitination and K48- and K63-linked polyubiquitination are the most abundant forms, multiple other types of ubiquitination exist which have distinct functional outcomes [6]. Mono-ubiquitination refers to the attachment of a single ubiquitin molecule to a target protein and serves as a signal for protein recognition, complex formation, or allosteric regulation. The addition of a single ubiquitin moiety during mono-ubiquitination can influence the localization, activity, or interaction of the modified protein within the cell. Poly-ubiquitination refers to the formation of a covalently linked ubiquitin molecule chain attached to a specific lysine residue of ubiquitin. When a protein is poly-ubiquitinated with K48-linked ubiquitin chains, it is recognized by the proteasome and targeted for degradation. Unlike K48-linked ubiquitin chains, K63-linked ubiquitin chains do not target proteins for degradation but enable context-specific functions of K63 ubiquitinated proteins in cellular signaling, intracellular trafficking, autophagy, and DNA damage responses [7,8]. K63-linked ubiquitin chains serve as scaffolds for protein–protein interactions, modulate enzyme activity, and regulate the localization and function of target proteins [9,10].

Deubiquitinating enzymes (DUBs) contribute to the regulation of a variety of biological processes, including proteasomal degradation of proteins, cell cycle regulation, histone modifications, transcriptional and translational control, protein trafficking, macro- and mitophagy, DNA damage response, epigenetic processes, and immune response signaling [5]. DUBs reverse the process of protein ubiquitination by selectively removing ubiquitin molecules or chains from proteins. Hence, DUBs are editors of the ubiquitin code and remove single ubiquitin molecules, entire ubiquitin chains, or ubiquitin branches from a ubiquitinated protein by cleaving ubiquitin substrate bonds and ubiquitin–ubiquitin peptide bonds [11,12] The approx. 100 putative DUBs identified so far in the human proteome are classified into five major families based on their structural and functional characteristics [13,14,15]. The ubiquitin-specific proteases (USPs) are the largest subclass of DUBs, with currently 54 members in humans [16]. USPs contain a conserved catalytic domain known as the ubiquitin-specific protease domain and exhibit specificity towards different types of ubiquitin linkages. Ubiquitin carboxy-terminal hydrolases (UCH) family members (four members) possess a distinct catalytic domain called the UCH domain. UCHL members are involved in the processing of ubiquitin precursors and the removal of ubiquitin from proteins [17,18,19]. DUBs of the ovarian tumor proteases (OTU) family (16 members) contain an ovarian tumor (OTU) domain and are involved in various cellular processes, including immune signaling, pathogen infection, and DNA damage response [20,21,22]. The Machado–Joseph disease proteases (MJD) family of DUB proteins (four members) possesses a Josephin domain, prefers K48/ K63 linkages, and is associated with neurodegenerative disorders, particularly Machado–Joseph disease [23]. The JAB1/MPN/Mov34 (JAMM) metalloenzyme family (16 members) of DUBs contains a metalloprotease domain and prefers targeting K63 ubiquitination sites. JAMM member CSN5 is a deNEDDylase [24,25,26]. The MINDY family is a recent DUB addition, with two of the four family members containing a “motif interacting with ubiquitin” (MIU) which assists in the enzymatic cleave of long K48 polyubiquitin chains [27]. Finally, a diverse group of ubiquitin-like proteases (ULPs) targets ubiquitin-like modifiers other than ubiquitin and comprises SENP (sentrin/ SUMO-specific protease), DeSI (deSUMOylating isopeptidase) families], and NEDD8 [28,29,30].

DUBs are essential for the dynamic and coordinated actions of the UPS and ensure proper functions of virtually all cellular processes, including the control of cellular levels of key regulatory transcription factors, growth factors, morphogens, cell cycle regulators, and the balance of factors regulating cell survival. The enzymatic removal of ubiquitin groups by DUBs is critical for reversible ubiquitination and the recycling of unbound ubiquitin to the UPS and ERAD (endoplasmic reticulum-(ER) associated degradation) pathways. Cellular DUB activity determines the coordinated regulation of both the UPS and ERAD pathways in a tissue region- and context-specific manner [31].

Extending throughout eukaryotic cells, the ER is the largest cellular organelle and composed of a series of sheet-like and tubular structures that form close contacts with other organelles, in particular the nucleus and mitochondria [32]. The ER can be subdivided into two types, the smooth and the rough ER. While the smooth ER facilitates lipid synthesis and hormone synthesis, the rough ER is the site of protein folding, modification, and quality control [32]. Signal peptides direct newly synthesized proteins to the ER lumen where ER-localized chaperones and enzymes facilitate protein folding and modifications. Correctly folded and processed proteins are then shuttled, via transport vesicles, to the Golgi apparatus and from there to their final destination [33]. The folding and modification of proteins is highly dependent on the maintenance of a stable ER environment. Exposure to stresses, such as oxygen and glucose deprivation or loss of ER calcium lowers protein folding efficiency, resulting in the accumulation of unfolded/misfolded proteins [34]. ER function can also be compromised by protein folding demands exceeding capacity. An example is viral infections where the capacity of the ER to facilitate protein folding is overwhelmed, giving rise to misfolded proteins [35]. Irrespective of the initiating stimulus, the buildup of misfolded/unfolded proteins is commonly referred to as ER stress. Cells combat ER stress by initiating an adaptive, highly conserved stress response referred to as the Unfolded Protein Response or UPR. The UPR is controlled by three ER-anchored transmembrane receptors, Inositol-requiring enzyme 1 (IRE1), protein kinase R (PKR)-like endoplasmic reticulum kinase (PERK), and activating transcription factor 6 (ATF6). These three ER-based receptors monitor ER health. In an unstressed setting, each of these stress sensors is held in an “off” position by binding their luminal domain to the ER chaperone Grp78 (BIP, HSPA5) [36,37]. Upon ER stress, Grp78 dissociates from IRE1, PERK, and ATF6, which stimulates their transition from inactive to active states [36,37]. Downstream signaling pathways orchestrated by IRE1, PERK, and ATF6 function in a co-operative, complementary manner to support the refolding of those proteins that can be refolded. Those proteins beyond repair are removed via the ER-associated degradation or ERAD pathway [34]. Ultimately, the objective of the UPR is to reduce ER stress, thereby restoring ER homeostasis. A functional UPS is a critical element of this cellular quality control system and DUBs are dynamically involved in this process.

Cancer cells frequently endure both external stressors (e.g., hypoxia and glucose deprivation) and internal stresses triggered by their high proliferation and metabolic rate. To thrive under such conditions and escape immune responses, cancer cells engage and coordinate adaptive responses, including UPS, ERAD, UPR, and DNA damage repair [38,39,40,41]. Although these responses may initially be engaged to aid cellular stress adaptation, cancer cells usurp and/or co-opt these pathways for their benefit in numerous ways. We recently identified distinct gene expression changes in ubiquitin ligases and ligase adaptors in different human brain tumors and subtypes [42,43]. Sustained UPR signaling has been reported in diverse cancers, including breast, prostate, and brain cancers, and emerging evidence links ERAD and UPR to an array of pro-tumorigenic processes, including angiogenesis, metastasis, and cancer stem cell expansion [38].

DUBs are an integral part of the UPS but their role in human brain tumors is incompletely understood. We gathered that understanding the role of DUBs in brain tumors could yield new therapeutic avenues. In the present study, we have analyzed the gene expression profiles of 99 human DUBs belonging to 7 subgroups listed in the HUGO (Human Genome Organization) classification. The objective of the current study was to comprehensively examine the differential gene expression of these DUBs in publicly available datasets of selected human neuronal system tumors, including pediatric craniopharyngioma (CPh), ependymoma (EPN), medulloblastoma (MB), adult brain tumors (astrocytoma (AS), oligodendroglioma (ODG), glioblastoma (GBM), and neuroblastoma (NBT) tumors arising from the developing sympathetic nervous system as the most common childhood extra-cranial neoplasm [44,45]. Some of the datasets had available gene expression data of non-tumor tissues for comparison, while other datasets had available age data for each subject. This allowed for plotting gene expression by tumor subgroup and by age for each DUB and enabled the comparison of gene expression in pediatric vs adult age groups. Adding novel insight, we sought to determine whether the expression of DUB genes was selective for specific CNS tumors, specific subgroups of CNS tumors, or specific age groups of subjects with these brain tumors. For those datasets with survival data, we determined whether DUB gene expression was statistically associated with survival. Top DUB hits identified in these bioinformatic screens were interrogated for their association with ERAD, UPR, and DNA repair. This is the first comprehensive report with a focus on DUB family members in selected pediatric and adult brain tumors, their relationship with ERAD, UPR, DNA damage repair pathways, and their suitability as potential biomarkers and therapeutic targets in defined CNS tumors. However, the function of some of the DUBs mentioned in the current report has previously been described in relation to CNS tumorigenesis [46,47].

Microarray datasets of CNS tumors available for the current study included those of glioma, ependymoma, medulloblastoma, craniopharyngioma, and neuroblastoma. Glioma originate from glial cells in the brain and include astrocytoma, glioblastoma, oligodendroglioma, and ependymoma. AS and GBM are considered to originate from astrocytes according to the WHO classification and astrocytoma can convert into GBM [48]. The hypothesis that GBM can originate from neural stem cells has also been proposed [49,50]. ODG originate from oligodendrocytes, while EPN derived from a type of glial cells called ependymocytes can be present intracranially and in the spinal cord, which affects treatment strategies [51,52]. MB are grouped by a consensus classification into four subgroups, the WNT group originating in the lower rhombic lip area of the hindbrain, the SHH group originating in the upper rhombic lip area [53], while MBs in Groups 3 and 4 arise from progenitor cells in the ventricular rhombic lip [54]. In the Cavalli dataset, each of the four consensus groups has been further characterized for subtype-specific molecular and clinical differences. A recent review has summarized the significance of molecular subtypes for the diagnosis and treatment of MB [55]. CPh are non-glial tumors originating in the hypothalamic and pituitary regions and are associated with remnants of Rathke’s pouch. Alomari et al. (2015) have presented a case of craniopharyngioma derived from Rathke’s cleft cyst and have reviewed the literature supporting the view that CPh originate from Rathke’s pouch cells [56]. NBT are sympathetic nervous system tumors originating from neural crest cells [57] and several excellent reviews discuss therapeutic strategies for NBT [58,59,60]. 

## 2. Methods

We utilized the human genome list of 99 deubiquitinases to determine the expression of these DUB genes in publicly available datasets of brain tumors. Differential expression of DUB genes was examined in these datasets made available in the R2 Genomics Analysis and Visualization platform (https://r2/amc/nl (accessed on 5 September 2023)). We used the AS, GBM, ODG, and a non-tumor group in the mixed glioma dataset of Sun et al. (Geo ID: GSE4290). Differentially expressed DUB genes were considered significant at *p* < 0.001 as determined by Analysis of Variance (ANOVA) through the R2 Genomics site and plotted using the Morpheus heatmap and cluster analysis program at the Broad Institute website (https://software.broadinstitute.org/morpheus (accessed on 5 September 2023)). DUB gene expression of classic, mesenchymal, and proneural GBM subtypes was examined in the TCGA GBM dataset (R2 ID: Tumor Glioblastoma TCGA 540). Survival data associated with differentially expressed DUB genes were examined in the French glioma data (GSE16022) [61]. The Pfister dataset (GSE64415) was used to examine differentially expressed DUB genes in ependymoma. A dataset of Donson (GSE94349) was used to examine the differential expression of DUB genes in CPh, while the datasets of Cavalli et al. [62] (GSE85217) and Weishaupt et al. [63] (GSE124814) were used to examine DUB gene expression in MB subgroups. The Cavalli dataset had extensive data on the age of subjects in the various subgroups, which we used to determine the age distribution for the most highly significant differentially expressed DUB genes. Those DUB genes statistically associated with the survival of MB patients were also determined in the Cavalli dataset. The Weishaupt data (Swartling dataset in the R2 Genomics database) allowed for a comparison of genes between MB tumor tissue and non-tumor tissue. The NBT dataset of Fischer [64] (GSE120572) was used to evaluate various treatment effects on DUB expression. The Cytoscape program was applied to identify gene ontology (GO) pathways, including ERAD, DNA repair, immune response pathways, and genes coding for differentially expressed DUBs that were associated with brain tumors.

## 3. Results

Cytoscape analysis of the 99 DUB genes (GO biological pathways) identified several functional groups of DUBs involved in the deubiquitination of lysine (K) residues K6/K11/K27/K29/K48/K63 ubiquitinated proteins. The most significant biological pathways, other than deubiquitination itself, included DNA repair, DNA methylation, the regulation of ER stress and ERAD pathway, death receptor signaling, and the regulation of immune and cytokine responses.

### 3.1. Adult Glioma Show Differential Expression of DUBs

In the Sun mixed glioma dataset, the expression of 51 of the 99 DUB genes (HUGO classification) was significantly different (*p* < 0.001) between the 4 groups in the dataset (non-tumor, AS, GBM, ODG). The heatmap (Figure 1) shows the gene expression profiles of the four groups. Table 1 shows the DUB genes that were most significantly different between the non-tumor group and each of the other three adult glioma groups. Seventeen of the ninety-nine DUB genes were differentially expressed (*p* < 0.001) between AS, ODG, and GBM (*USP46*, *USP54*, *ZRANB1*, *USP1*, *OTUD7A*, *TNFAIP3*, *USP27x*, *USP30*, *EIF3H*, *USP49*, *OTUD1*, *USP11*, *OTUB1*, *CYLD*, *USP12*, *USP2*, *USP47*, in order of *p* value as determined by ANOVA).

Next, we asked whether the differences in DUB gene expression between AS and GBM were related to the progression from AS to GBM [48], which is associated with several changes in the transcriptome [65]. ANOVA showed that the expression of two DUB genes, *USP46* and *ZRANB1*, differed at a high level of significance (*p* < 0.001) between the AS and GBM groups (Figure 2). *USP46* was among the top 100 of all differentially expressed genes between the AS and GBM groups in the Sun mixed glioma dataset. Of all 18,896 genes in the French dataset, *ZRANB1* and *USP46* expression were ranked 3rd and 2725th, respectively, when associated with survival. ZRANB1 belongs to the OTU class of DUBs and has been reported as an EZH2 (enhancer of zeste homolog 2) DUB [66]. EZH2 inhibitors are currently tested for cancer therapy and brain permeable derivatives may offer new avenues in the treatment of brain tumors [67,68].

### 3.2. Chromosome 10 and DUB Expression in Astrocytic glioma

Table 2 shows DUB gene expression related to survival in the French dataset of glioma subjects. Expression of *ZRANB1*, a gene located on chromosome 10, showed the most significant differences in Kaplan Meier survival curves. In addition to *ZRANB1*, several other differentially expressed DUB genes are also located on chromosome 10. The HUGO list of DUB genes includes two DUBs located on the p arm of chromosome 10 (*OTUD1* and *MINDY3*/*FAM188A*) and three DUBs that are located on the q arm of chromosome 10 (*ZRANB1*, *USP54*, and *STAMBPL1*). Expression of *ZRANB1* and *USP54* was decreased in the GBM group compared to the AS and non-tumor groups.

The loss of chromosome 10 in primary GBM or loss of the q arm of chromosome 10 in secondary GBM is a common finding [69,70]. This has led to the hypothesis that the loss of one or more tumor suppressor genes on the q arm of chromosome 10 may contribute to GBM development. The most significant differentially expressed pathway between the AS and GBM group in the Sun dataset was the “*D-glutamine and D-glutamate*” pathway, which was represented by differential expression of two genes, *GLUD1* and *GLUD2* (glutamate dehydrogenase 1 and 2). *GLUD1* is located on chromosome 10 and codes for the mitochondrial matrix enzyme glutamate dehydrogenase 1. *GLUD1* and *GLUD2* expression was lowered to 52.7% and 52.2% compared to the NT and AS groups, respectively. While a role for specific DUBs in the regulation of *GLUD1* has not been established, among all 99 DUB genes we observed the highest correlations of *GLUD1* expression with *USP46* (r = 0.74, *p* = 1.74 × 10^−30^) and *ZRANB1* (r = 0.70, *p* = 1.01 × 10^−26^). This suggests a possible new role for USP46 and/or ZRANB1 in regulating *GLUD1* in astrocytic glioma (AS and GBM).

### 3.3. Differential Expression of DUBs in GBM Subtypes

The differential expression of DUB genes was determined in three GBM subtypes from a TCGA dataset in the R2 genomics platform. *TNFAIP3* showed the most significant difference (*p* = 7.68 × 10^−09^) with elevated expression in the mesenchymal GBM subtype (Figure 3). The DUBs with the most significantly elevated gene expression in the proneural GBM group were *USP11*, *USP22*, and *USP7*.

### 3.4. Relationship between DUB Expression and Survival (French Dataset)

Survival data were not available in the Sun mixed glioma dataset. Hence, we used the glioma dataset (GSE16011) of French et al. [61] (N = 284) to determine survival data associated with DUB gene expression. Kaplan Meier curves showed that the expression of 23 DUB genes was significantly associated (*p* < 0.001) with survival in glioma patients. Table 2 lists the Chi square, p values, and hazard ratios for these DUB genes. The most significantly downregulated DUB gene, *ZRANB1* (as would be expected with loss of chromosome 10 or its q arm), was associated with worse survival (Table 2). 

### 3.5. Role of DUBs in ER Stress and ERAD Signaling in Glioma

Employing the R2 genomics platform to query the 99 DUBs for their association with the GO category of *Regulation of the ERAD pathway*, we identified 3 DUB genes in this category as differentially expressed in the glioma dataset: *USP14*, *USP19*, and *USP25* (Table 3). It should be noted, however, that only *USP14* was among the most significant in Table 1. In the French dataset, high *USP14* expression was associated with worse survival (Table 2). The role of USP14 and USP19 proteins in ER stress has been illustrated in a review by Qu et al. [31]. USP19 is reported to inhibit the unfolded protein response [71] and to deubiquitinate the E3 ligase HRD1 [72], a component of the ERAD pathway. USP14 binds to IRE1 and is reported to be an inhibitor of the ERAD pathway [73]. USP25 deubiquitinates selected ERAD substrates [74]. Another DUB reported by Qu et al. to regulate ER stress-induced apoptosis is BAP1 [31]. Differential *BAP1* gene expression is also shown in Table 3.

### 3.6. DUBs in the Regulation of Immune Responses in Glioma

Among the 99 DUB genes, 4 differentially expressed genes were identified in the Sun glioma dataset that were associated with the GO category of *Regulation of the immune response*. This included *OTUD7A*, F = 69.909, *p* = 1.31 × 10^−29^, *CYLD*, F = 50.614, *p* = 1.69 × 10^−23^, *TNFAIP3*, F = 11.84, *p* = 4.34 × 10^−07^, and *USP18*, F = 8.14, *p* = 4.21 × 10^−05^ (Figure 4). Notably, the expression of *OTUD7A* was not only the most significantly different among the DUBs in the GO category of *Regulation of immune response* but was also the most significant of any of the 512 genes in this category in the Sun dataset. *OTUD7A* expression was significantly decreased in AS, GBM, and ODG (Figure 4), as was *OTUB1* in all three types of gliomas compared to non-tumor tissues in the Sun dataset (Table 1). OTUB1 deubiquitinase function was recently associated with the regulation of immune responses and contributes to immunosuppression in cancers via the programmed death ligand 1 (PD-L1) protein [75]. Decreased *OTUB7A* and *OTUB1* gene expression may both affect immune responses and DNA damage repair functions (see below) in glioma [75,76].

The *OTUD7A* gene is located on chromosome 15q13.3. Microdeletion of this chromosomal region results in abnormalities of neuronal development [77,78]. CYLD is a tumor suppressor that contributes to the regulation of NF-κB [79]. TNFAIP3 plays a role in several aspects of the immune response, including the regulation of NF-κB and the regulation of inflammation [80]. *TNFAIP3* deletions have been associated with Epstein–Barr viral infection in lymphomas [81]. USP18 regulates interferon signaling by binding to one of its receptors (IFNAR2) [82].

### 3.7. DUBs and DNA Repair in Glioma

Ten of the ninety-nine DUB genes analyzed in the Sun dataset were differentially expressed genes associated with the GO category of *DNA repair*: *OTUB1*, *UCHL5*, *USP3*, *USP1*, *USP51*, *COPS5*, *COPS6*, *USP10*, *USP47*, and *USP43* (in order of significance in ANOVA). The expression of *OTUB1* and *UCHL5* was significantly decreased (*p* < 0.001) in AS, GBM, and ODG compared to non-tumor tissue, while the expression of *USP3* was significantly elevated (*p* < 0.001) (Figure 5). Expression of *USP1* was markedly increased in GBM and, to a lesser extent, in AS compared to the non-tumor group. Several DUBs, including *USP1*, *OTUB1*, *UCHL5*, and *USP47*, have been included in a list of 16 DUBs reported to be involved in specific DNA damage repair pathways [83]. OTUD7A (Cezanne2) has also recently been reported to contribute to the DNA damage response in double-strand break (DSB) repair [84] and expression of *OTUD7A* was substantially reduced in gliomas compared to non-tumor brain tissue (Figure 4). While linked to several DNA repair pathways [83], *USP7* and *USP24* expression was not significantly different among glioma groups or between glioma and the non-tumor control group in the Sun dataset, suggesting that these two DUBs may not be critical factors in DNA damage repair pathways in glioma.

### 3.8. DUBs in Ependymoma

The heatmap and cluster analysis shown in Figure 6 illustrate clusters of DUB gene expression that differ significantly between the EPN subgroups of the Pfister dataset. Of all molecular subgroups in this dataset, the largest groups were the posterior fossa groups, Pf_Epn_a (N = 72) and Pf_Epn_b (N = 39) followed by the supratentorial group (St_Epn_Rela) (N = 49). Among all 99 DUBs, *USP30* and *STAMBPL1* were most significantly different in these 3 EPN subgroups. *USP30* expression most significantly distinguished Pf_Epn_a and Pf_Epn_b, whereas *STAMBPL1* expression was decreased compared to the other subgroups (Figure 7).

The UPS30 protein is located on the mitochondrial outer membrane [85] and serves as an inhibitor of mitophagy [86,87] by blocking the action of the E3 ligase PARKIN [88]. STAMBPL1 is a K63-specific DUB reported to have higher expression in cancer tissue than in adjacent control tissues [89,90].

The heatmap of DUB expression in EPN subgroups (Figure 6) showed a cluster with relatively elevated DUB expression in the St_Rela subgroup (red squares in heatmap) and a cluster in which DUB expression was relatively decreased (blue squares in heatmap). Cytoscape analysis of GO biological pathways identified several DUBs associated with *histone deubiquitination* in both clusters. This included the upregulated expression of *BAP1*, *USP25*, *USP3*, and *USP49* (red squares), and downregulated expression of *USP16*, *USP21*, *USP22*, and *USP51* (blue squares) (Figure 6). Cytoscape Reactome pathway analysis of differentially expressed DUB genes in Figure 6 identified the expression of three DUB genes, *CYLD*, *USP2*, and *USP21*, to be associated with the TRAF2:RIP1 complex in tumor necrosis factor receptor (TNFR) signaling and apoptosis. The UPS2 protein has been labeled a “master regulator of apoptosis” since USP2 can remove ubiquitin chains from RIP1 and TRAF2, regulate TNF-TNFR1-mediated cell death, and upregulate the transcription of IkBα [91]. Like USP2, USP21 was also reported to deubiquitinate RIP1 [92] and the selective USP21 inhibitor compound BAY-805 may have therapeutic potential in cancer [93]. Both CYLD and TNFAIP3 have been shown to also contribute to the regulation of NF-κB [79,81]. Notably, the supratentorial molecular subgroup St_Se of EPN was unique in that it showed increased expression of a cluster of four DUB genes, *CYLD*, *USP46*, *USP53*, and *USP32* (Figure 6). This may be considered a new gene signature for this WHO grade I subependymoma (Se) subgroup [94]. Among the significant differentially expressed DUBs in the EPN dataset, five DUB genes were associated with the *Regulation of ERAD pathway*, including *ATXN3*, *USP3*, *USP14*, *USP25*, and OTUD2/*YOD1*. Since the Pfister ependymoma dataset did not include non-tumor subjects or survival data, these comparisons were not possible for the DUB genes.

### 3.9. DUBs in Craniopharyngioma

Of the 99 DUB genes analyzed, 39 DUBs were differentially expressed between normal brain and CPh. Other than DUB activity itself, *histone deubiquitination* (USP3, USP7, USP16) was the most significant GO pathway identified by Cytoscape.

*USP13* (F = 24.25, *p* = 1.00 × 10^−05^) and *USP14* (F = 11.88, *p* = 1.17 × 10^−03^) expression were both decreased in CPh compared to non-tumor tissue (Figure 8). Of note, differential expression of *USP14* was also observed in mixed gliomas and in EPN. USP14 protein was reported to be an inhibitor of the ERAD pathway by binding to IRE1a and inhibiting the phosphorylation of this ER stress-activated kinase [95].

### 3.10. DUB Regulation of Immune Response in Craniopharyngioma

In the dataset of CPh, the DUB genes *TNFAIP3*, *OTUD7A*, and *CYLD* were identified by Cytoscape to be associated with the *Regulation of immune response* pathway. *TNFAIP* expression was upregulated about five-fold (F = 78.84, *p* = 9.93 × 10^−12^) compared to non-tumor tissue. TNFAIP3 has been identified as a druggable target for melanoma in mice [96] and in inflammatory lung disease [97]. Expression of *OTUD7A* (aka Cezanne2) in CPh was significantly downregulated to 6.8% of normal non-tumor values (F = 81.52, *p* = 5.34 × 10^−12^). Located on chromosome 15, *OTUD7A* is one of six genes that contribute to the 15q13.3 microdeletion syndrome, which is associated with neurodevelopmental and psychiatric disorders [98,99]. Whether OTUD7A protein may qualify as a tumor suppressor for the development of CPh tumors (and gliomas, see above), as these data may suggest, requires further studies. Expression of the DUB and tumor suppressor CYLD was significantly reduced (>50%) in CPh compared to normal brain tissue. CYLD is an inhibitor of the immune response and NF-κB signaling [100,101,102].

### 3.11. DUBs and Medulloblastoma

Seventy-eight of the ninety-nine DUB genes were differentially expressed (*p* < 0.001) among the four subgroups of MB. Table 4 shows the top ten DUB genes of each MB subgroup most significantly different from non-tumor tissues in the Swartling dataset. The heatmap in Figure 9 illustrates the distribution of DUB expression among the four MB subgroups (Group 3, Group 4, SHH, and WNT) in the Cavalli dataset. The most significant differentially expressed DUB gene by subgroups in the Cavalli dataset was *USP2* (F = 271.00, *p* = 1.57 × 10^−119^) (Figure 9), which was upregulated in Group 3 MB compared to the other subgroups. USP2 protein removes ubiquitin from several proteins, including the E3 ubiquitin ligase MDM2 [103], cyclin D (CCND1) [104], and the circadian clock protein PER1 [105]. While the Cavalli MB dataset does not include non-tumor controls, the Donson dataset showed that *USP2* expression was elevated compared to the non-tumor brain. This was confirmed in the large meta-analysis of Weishaupt et al. [63] available in the R2 genomics site as the Swartling dataset, which identified *USP2* as the most significant differentially expressed gene in the five groups (non-tumor, Group 3, Group 4, SHH, Wnt). The role of USP2 in the deubiquitination of clock proteins regulating the circadian rhythm of pathways is well documented [106,107]. USP2 may function as an oncogene in breast and gastric cancer by inhibiting autophagy and this DUB has been proposed as a therapeutic target in breast cancer [108,109]. *USP2* expression was also differentially expressed according to age groups (F = 13.73, *p* = 1.01 × 10^−08^). The age distribution for the four MB subgroups showed elevated *USP2* expression primarily in infants and children (Figure 10) and high expression of *USP2* was associated with poor survival (Table 4). USP2 may be a lucrative therapeutic target in patients with Group 3 MB.

### 3.12. Survival in Medulloblastoma Subgroups and DUB Expression

Overall survival in the Cavalli dataset, as determined by the R2 genomics platform, was best in the Wnt group, worst in Group 3, and intermediate in SHH and Group 4, confirming previously determined survival times [62]. DUB genes that were significantly associated with survival (*p* < 0.01, for Kaplan Meier curves) are listed in Table 5 and includes DUB genes not differentially expressed between subgroups. DUB genes with the most significant Kaplan Meier curves (high vs low) and hazard ratios included *VCPIP1*, *USP49*, and *USP2*. High expression of these genes was associated with worse survival (Table 4). While high expression of *VCPIP1* was associated with worse survival in the Cavalli dataset, expression of *VCPIP1* in none of the four MB groups was significantly higher than in non-tumor tissues in the Swartling dataset. High expression of *USP49* was observed in infants and young children in Groups 3 and 4, whereas high expression of *USP2* was primarily observed in infants and young children in Group 3 (Figure 10).

### 3.13. Medulloblastoma DUBs and ERAD

Differential expression of several DUB genes (*USP13*, *USP14*, USP25, USP19, *OTUD2* (alias *YOD1*)) in the Cavalli dataset was associated with the GO category of *Regulation of ERAD pathway*. This list of genes includes three DUBs (*USP14*, *USP19*, *USP25*) shared with the list of differentially expressed ERAD genes in glioma (Table 3). *USP25* was the only ERAD-associated DUB gene among the top ten DUB genes in the SHH MB group (Table 4). Our data implicate both *USP25* and *USP13* with ERAD in the SHH group of MB (Figure 11). In the Swartling dataset, the expression of *USP25* was downregulated in the SHH group compared to non-tumor controls (t = 14.37, *p* = 3.17 × 10^−41^), while the expression of *USP13* was elevated compared to non-tumor tissues (t = 11.36, *p* = 1.41 × 10^−27^). Expression of *USP14*, but not *USP19* and *OTUD2*, was also reduced in the SHH group versus non-tumor tissues, but at a much lower level of significance (t = 3.35, *p* = 8.64 × 10^−04^) 

### 3.14. Medulloblastoma DUBs and the Immune Response

Five differentially expressed DUB genes were associated with the GO category of *Regulation of immune response* in the Cavalli dataset. This included DUB genes *CYLD* (F = 141.44, *p* = 8.53 × 10^−73^), *PSMD14* (F = 58.20, *p* = 7.15 × 10^−34^), *OTUD7A* (F = 66.35, *p* = 4.14 × 10^−38^), *USP18* (F = 37.27, *p* = 1.78 × 10^−22^), and *PSMD7* (F= 16.56, *p* = 1.96 × 10^−10^). Table 6 shows the genes that were significantly elevated or decreased compared to non-tumor samples in the Swartling dataset.

CYLD is an inhibitor of the immune response, alters NF-κB signaling, and affects the development and Th2 conversion of Treg cells [101,102,110]. PSMD14 and PSMD7 are DUB components of the proteasome and PSMD14 is a druggable target that specifically deubiquitinates at K63 and suppresses autophagy by affecting vesicular retrograde transport from the Golgi to the ER [111,112]. OTUD7A contributes to neuronal development [77,78] and USP18 regulates the immune response by binding to the interferon receptor IFNAR2 [82].

### 3.15. Medulloblastoma DUBs and DNA Damage Repair

Twelve of the ninety-nine DUB genes were identified as differentially expressed in MB compared to non-tumor tissue (*p* < 0.0001) for the GO category of *DNA repair* in the Swartling dataset, including *USP1*, *OTUB1*, *UCHL5*, *USP7*, and *PSMD14* (aka *POH1*) (Table 7). The DUB proteins USP1, OTUB1, UCHL5, USP7, and PSMD14 were reported to contribute to double-strand break repair, USP1 to Fanconi anemia pathway, USP1 and USP7 to translesion repair, USP7 and USP47 to base excision repair, and USP7 and USP45 to nucleotide excision repair [83]. In addition, the USP28 protein was also found to contribute to the DNA damage response [113]. A highly significant reduction in *USP28* expression in Group 3, Group 4, and Wnt MB groups (Table 7) may point to an impaired DNA damage response in these MB groups. *PSMD14* was the most significantly upregulated DUB gene in Group 3 MB (Table 7) [114]. A subtype-specific analysis revealed that *PSMD14* over-expression was limited to selected subtypes of Group 3 (Group 3 beta and gamma) and Group 4 (Group 4 alpha).

### 3.16. DUBs in Neuroblastoma (Fischer Dataset)

The dataset of Fischer on NBT allowed the examination of DUB genes in patients with various treatments. Intensive chemotherapy of NBT was associated with increased expression of selected DUB genes and a decrease in several other DUB members compared to the observation group. Intriguingly, limited chemotherapy or surgery had no significant effect on the expression of DUB genes compared to the observation group alone (Figure 12).

Patients receiving intensive chemotherapy of their NBT showed significantly reduced tumor tissue expression of *USP24*, *USP34*, *MINDY2*, *USP8*, *JOSD1*, *USP52*, and *USP12* when compared to the observation group. The most significant differences in DUB expression between limited and intensive chemotherapy included *USP24*, *JOSD1*, and *MINDY2* (Figure 13). Notably, there were many other genes in the Fischer dataset that showed differential expression between the limited and intensive chemotherapy groups, the most significant of them being *MDGA1* (F = 128.15, *p* = 4.24 × 10^−21^) with approximately a four-fold difference. *MDGA1* is expressed mainly in neurons and astrocytes of the brain.

## 4. Discussion

The complex functional roles of DUBs in tumor biology are gradually emerging [115]. Here, we present the first comprehensive gene expression profiling of 99 DUB family members in 6 different brain tumor entities that span different molecular subtypes and age groups. We also included gene expression data from different treatment groups of NBT, the most common extracranial sympathetic nervous system tumor in children, to better understand the emerging role of DUBs in pediatric and adult CNS tumors and the effect of treatment on DUB expression. While we observed pronounced gene expression changes for several DUBs, for brevity, we will only discuss those selected DUB members with the highest differential expression. Wherever possible, we have focused on known brain-related functions for these DUBs, with a particular emphasis on clinically relevant pathomechanisms, such as the ERAD pathway, immune system, and DNA damage repair.

GBM displayed a distinct downregulation of *USP46* and *ZRANB1*. Ubiquitously expressed throughout the mammalian brain, USP46 is involved in the formation of synapses and neuronal morphogenesis by regulating both excitatory and inhibitory synaptic transmission [116]. By deubiquitinating K63 ubiquitinated glutaminergic AMPARs (α-amino-3-hydroxy-5methyl-4-isoxazolepropionic acid) receptors GluA1 and GluA2, which are considered to mediate most of the excitatory synaptic transmission in the brain, USP46 upregulates the intracellular trafficking, cell surface density, and signal intensity of AMPARs [116,117]. These receptors are critical for perivascular brain invasion, promote plasticity and growth of GBM, and coincide with poor prognosis [118,119,120]. USP46 also interferes with the neuronal activity-dependent ubiquitination and trafficking of GABA_A_ receptors. Loss of USP46 coincides with reduced expression of glutamic acid decarboxylase (GAD67) which synthesizes GABA [121]. Recently, higher expression of the non-coding (nc)RNA USP46-AS1 has been linked to increased overall survival in glioma [122]. It is tempting to speculate that the marked reduction in USP46 gene expression in GBM, but not AS, coincides with the acquisition of altered receptor density in the plasma membrane and synaptic activity during dedifferentiation from high-grade AS to GBM. GABA_A_ receptor activity was reported to inhibit glioma growth and the lowest levels of GABA_A_ receptors were reported in GBM compared to lower-grade glioma [123,124].

While the role of ZRANB1 (zinc finger RANBP2-type containing 1, TRABID) in glioma is still unclear, it is likely multifactorial in nature. In breast cancer, this K29- and K33-specific DUB binds, deubiquitinates, and stabilizes the enhancer of zeste homologue (EZH2) catalytic component of the gene silencing Polycomb repressive complex 2 (PRC2) to promote growth, resulting in poor prognosis [66]. USP1 in glioma [125], as well as USP7 and USP34, also converge on EZH2 to promote tumorigenesiss [126]. Tight regulation of ZRANB1 expression is critical in glioma. Reduced expression of ZRANB1 may confer a survival advantage to GBM by reducing UPR through the recruitment of p62 to K33-ubiquitated protein aggregates for autophagic removal [127]. However, in solid tumors, lower ZRANB1 levels coincide with epigenetic regulation that promotes interferon and inflammatory immune cell responses in the tumor microenvironment [128,129,130]. Lower ZRANB1 levels also attenuate the deubiquitination of K29-linked polyubiquitinated 53BP1. Proteasomal removal of this DNA repair factor mitigates genomic instability by preventing 53BP1 from blocking homologous recombination repair at double-strand DNA breaks [131]. Further studies are needed to establish the role of ZRANB1 in GBM.

Selected GBM, EPN, CPh, and MB subtypes showed distinct expression of specific DUB genes. Among 10 differentially expressed DUBs in the 3 GBM subtypes, only *TNFAIP3*, aka A20, was upregulated in mesenchymal GBM. Possessing both DUB and E3-ligase activities [132,133], TNFAIP3 is an important player in a diverse array of diseases [134] and a key negative regulator of NFkB signaling downstream of TNF receptors, interleukin 1 receptor (IL-1R), pathogen recognition receptors (PRRs), NOD-like receptors (NLRs), T- and B-cell receptors, and CD40 [135,136]. TNFAIP3 regulates glioma stem cell survival, increases resistance to alkylating agents, and is considered a poor prognostic marker in GBM [137,138]. While *TNFAIP3* upregulation was a unique mesenchymal feature among GBM subtypes, DUB genes significantly associated with immune cell functions were identified in other GBM subtypes, the St_se subgroup of EPN, MB, and in CPh. This included *TNFAIP3*, *CYLD* (*Cylindromatosis*), another negative regulator of NF-κB signaling [102], and the critical neurodevelopmental factor and putative tumor suppressor *OTUD7A*/*Cezanne-2* [78]. These data suggest a redundant role for several DUBs in targeting NF-κB signaling as a mechanism to regulate inflammatory and immune responses in intra- and extracranial nervous system brain tumors.

Among the four MB subgroups, we identified *USP2* to be selectively upregulated primarily in infants and children within Group 3 MB (Figure 9). Group 3 MB frequently have elevated MYC levels due to MYC overexpression or MYC gene amplifications and these patients have the worst prognosis of all MB groups with less than 50% survival [139,140]. In the Cavalli dataset, MYC expression was elevated most in the Group 3 gamma subtype. As may be expected, USP2 DUB functions target a wide range of interconnected pathways in a tissue-specific manner [141]. Relevant USP2 functions in tumorigenesis target the metabolic (e.g., fatty acids) and p53 pathways, EMT, cell cycle control, and maintenance of genome stability [141]. High USP2 levels resulted in the downregulation of several miRs, including *MYC*-targeting miR-34b/c, which resulted in the deubiquitination of MDM2 and elevated MYC levels with subsequent p53 inactivation in prostate cancer cells [142]. Hence, it is conceivable that higher USP2 expression may contribute to higher MYC protein levels in Group 3 MB patients.

Emerging research is starting to unravel the complex and clinically relevant relationships between UPR, ER, and DNA stress signaling, chronic inflammation, and immune responses in primary brain tumors and their microenvironment [143,144,145,146]. We identified a selected group of DUBs (*USP13*, *USP14*, *USP19*, *USP25*, *OTUD2*/*YOD1*) associated with the *Regulation of ERAD pathway* across several adult and pediatric primary brain tumors (GBM, EPN, CPh, MB). A recent TCGA-based gene expression profiling interactive analysis (GEPIA; http://gepia.cancer-pku.cn/ (accessed on 25 August 2023)) of low-grade glioma and GBM identified lower expression of all but one (*USP25*) of these USP Dub members in GBM [147]. There was a strong correlation between higher expression of *USP14* and worse prognosis in GBM patients [147]. In addition to its roles in the ER [73], USP14 (and UCH37) engages in polyubiquitin chain trimming which can delay proteasomal degradation by weakening the affinity of ubiquitin chains with ubiquitin-binding receptors at the proteasome [148]. Hence, USP14 has been targeted with a small molecule inhibitor [149] or selected USP14 aptamers [150] to enhance proteasomal activity and degradation of proteotoxicity. Although USP14 downregulation in several tumor types was shown to reduce tumor burden in mice, data are lacking for brain tumors [151,152]. OTUD2/YOD1 is another DUB with regulator functions in the ERAD pathway and is linked to injury-induced ER stress responses [153,154]. This includes a regulatory role of the inflammatory cytokine IL1 and p62 NFkB signaling axis through interaction with the E3 ligase TRAF6 [155], which may contribute to OTUD2/YOD1 deubiquitinating activity in attenuating neurogenic proteotoxicity [156]. In glioma, OTUD2/YOD1 has been identified as a target of miR-190a-3p. Blocking miR-190a-3p or the overexpression of its target OTUD2/YOD1 attenuated the proliferation and migration of glioma cells [157]. While the underlying mechanism is currently unknown, YAP and TAZ, the transcriptional coactivators and effectors of the Hippo signaling pathway, have been identified as downstream targets of an miR21-OTUD2-YAP/TAZ axis in hepatocellular carcinoma [158], thus, potentially linking OTUD2/YOD1 to glioma stem cell maintenance and proliferation [159].

Among the selected DUBs significantly linked to the DNA damage repair pathway, the glioma and MB datasets shared several DUBs, including *USP1*, *USP47*, *UCHL5*, and *OTUD1*, which cover five major DNA damage repair pathways (BER, NER, FA, TLS, DSB) [83]. USP1 targets FANCD2/FANCI to regulate the Fanconi anemia pathway (FA) [160] and, together with USP7, targets translesional DNA repair (TLS) [161,162]. USP7, a DDR-associated DUB exclusively altered in MB, and USP47 target the base excision repair pathway (BER) [163,164], while USP7 and USP45 have regulatory roles in nucleotide excision repair (NER) [165,166]. *UCHL5* and *OTUD1* were reported to increase or decrease double-strand break repair (DSB), respectively [167,168]. Expressed among the top genes in Group 3 MB and highly significantly associated with poor survival in MB Groups 3 and 4 (Table 4 and Table 5), *PSMD14* (aka *POH1*) was also significantly associated with immune responses and DNA repair, particularly in MB Groups 3 and 4 (Table 6 and Table 7). PSMD14 was shown to fortify tumor cells against DNA-damaging drugs by promoting a switch from error-prone non-homologous end-joining to homologous recombination [169,170]. This identifies proteasomal PSMD14 as a key DUB in regulating ubiquitin conjugation in response to DNA damage and exemplifies the intricate relationships between the proteasome and DNA damage responses.

Changes in DUB expression also occurred during the treatment of extracranial NBT sympathetic nervous tumors. Our analysis of a dataset from NBT undergoing different treatment options identified significantly reduced expression of seven (*USP24*, *USP34*, *MINDY2*, *USP8*, *JOSD1*, *USP52*, *USP12*) DUBs in the treated versus non-treated NBT group. Intriguingly, *USP24*, *JOSD1*, and *MINDY2* showed the most significant downregulation during intensive versus limited chemotherapy (Figure 12). USP24 has recently been identified as a novel tumor suppressor in NBT that targets collapsin response mediator protein 2 (CRMP2), which promotes axon growth, guidance, and neuronal polarity but also affects T cell polarization and migration [171,172]. Deubiquitination of CRMP2 by USP24 ensured proper spindle pole assembly and blocked chromosomal instability and aneuploidy observed upon USP24 knockdown in NBT [173]. A glimpse into possible additional cellular strategies in response to intensive treatment regimes in NBT cells comes from findings that USP24 downregulation increases autophagy flux in cells [174]. The biological roles of Machado–Joseph DUB member JOSD1 are complex [23]. While data on JOSD1 in NBT are lacking, JOSD1 can deubiquitinate and stabilize Snail protein to promote EMT and tissue invasion of lung cancer cells [175]. A small molecule inhibitor of JOSD1 was shown to induce cell death of JAK2-V617F-positive primary acute myeloid leukemia (AML) cells [176]. Downregulation of JOSD1 in treated NBT may also affect regulatory mechanisms of interferon-1 mediated inflammatory cytokine responses [177]. The role of the evolutionarily conserved MINDY1/2 family of DUBs in brain tumors is currently unknown. However, MINDY1 DUB activity promotes the proliferation of bladder cancer cells by stabilizing MINDY1 interaction partner YAP protein and critical transcriptional regulator of the Hippo pathway. YAP overexpression in MINDY1-depleted cells was able to rescue this proliferation [178]. In human breast cancer cells, MINDY1 stabilizes estrogen receptor alpha (ERa) and promotes Era-mediated proliferation [179].

## 5. Summary and Conclusions

In the current study, we have identified DUB coding genes and biological pathways that are statistically associated with CNS tumors. Several DUBs were specific for a particular CNS tumor or subgroup/subtype and others were common to two or more tumors. The *histone deubiquitination* GO pathway was over-represented in glioma, EPN, MB, CPh, and NBT datasets. The *DNA synthesis in DNA repair* and *regulation of ERAD pathways* were over-represented in DUB transcriptomes of MB and EPN, while various aspects of the immune response were associated with differential expression of DUBs in CNS tumors. For datasets that included survival data (mixed glioma and MB datasets), we identified DUB genes associated with significant hazard ratios. In conclusion, the role of DUBs as relevant modulators of cellular and immunoregulatory pathways in brain tumors is evolving. Selective DUB targeting strategies may provide important synergistic therapeutic potential in the future.

## Figures and Tables

**Figure 1 biomolecules-13-01503-f001:**
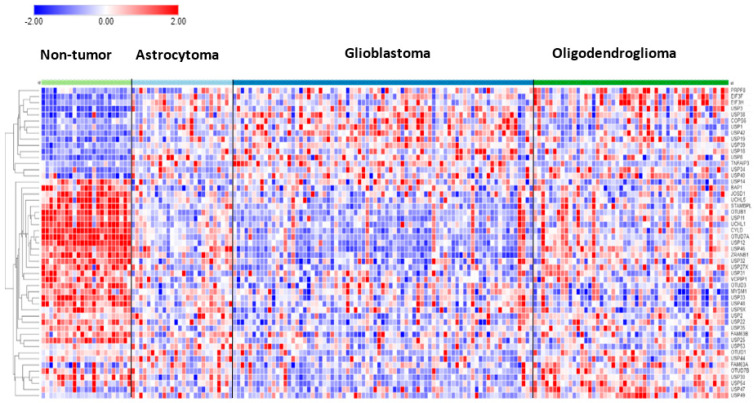
Heatmap and cluster analysis of differentially expressed DUBs in gliomas. Non-tumor, N = 23; Astrocytoma, N = 26; Glioblastoma, N = 77; Oligodendroglioma, N = 50.

**Figure 2 biomolecules-13-01503-f002:**
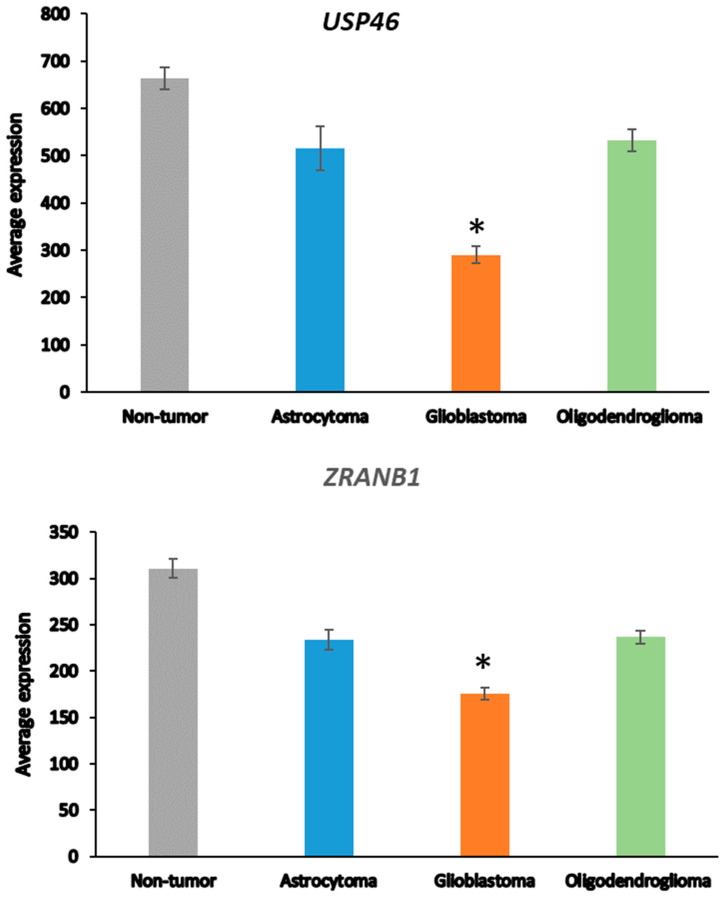
Differential expression of *USP46* and *ZRANB1* in adult glioma groups. *USP46* (F = 42.34, *p* = 1.53 × 10^−20^); *ZRANB1* (F = 45.17, *p* = 1.40 × 10^−21^) in the Sun dataset; Non-tumor N = 23, Astrocytoma, N = 26, Glioblastoma, N = 77, Oligodendroglioma, N = 50). * indicates significant difference between GBM and AS, *p* < 0.0001.

**Figure 3 biomolecules-13-01503-f003:**
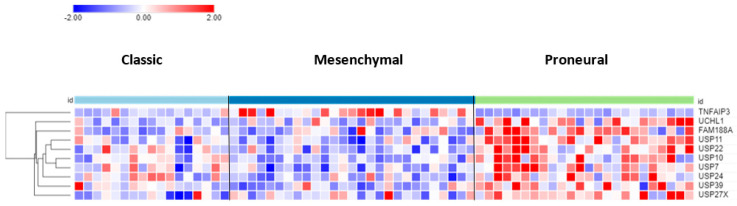
Heatmap and cluster analysis of DUB genes differentially expressed in GBM subtypes. On average, *TNFAIP3* expression was higher in the GBM mesenchymal group, whereas the expression of the other 9 genes was higher in the proneural group. Classic, N = 17; Mesenchymal, N = 27; Proneural, N = 24).

**Figure 4 biomolecules-13-01503-f004:**
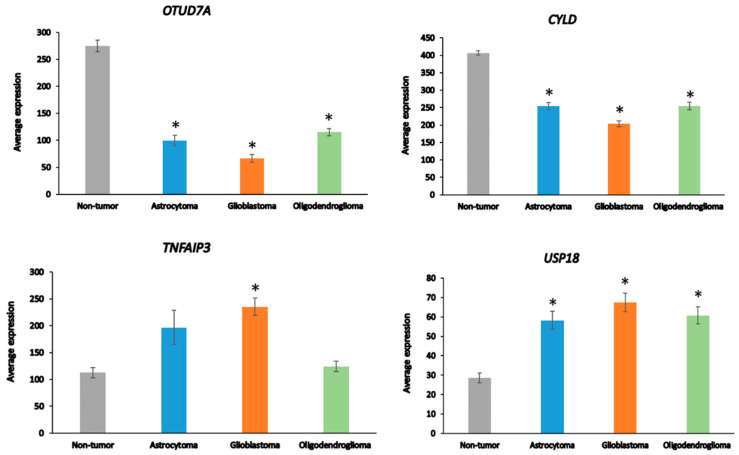
Differentially expressed DUBs identified under the GO category of *Regulation of immune response* in different glioma and non-tumor brain. * *p* < 0.001 compared to non-tumor group.

**Figure 5 biomolecules-13-01503-f005:**
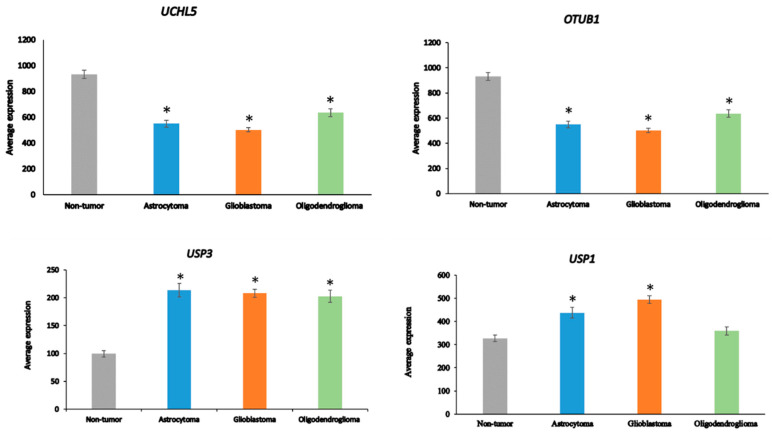
Differentially expressed DUB genes in the GO category of DNA repair. *UCHL5*, F = 21.83, *p* = 4.97 × 10^−12^; *OTUB1*, F = 42.71, *p* = 1.12 × 10^−20^; *USP3*, F = 19.33, *p* = 7.56 × 10^−11^; USP1, F = 16.99, *p* = 1.04 × 10^−09^. * *p* < 0.001 compared to non-tumor group by t-test.

**Figure 6 biomolecules-13-01503-f006:**
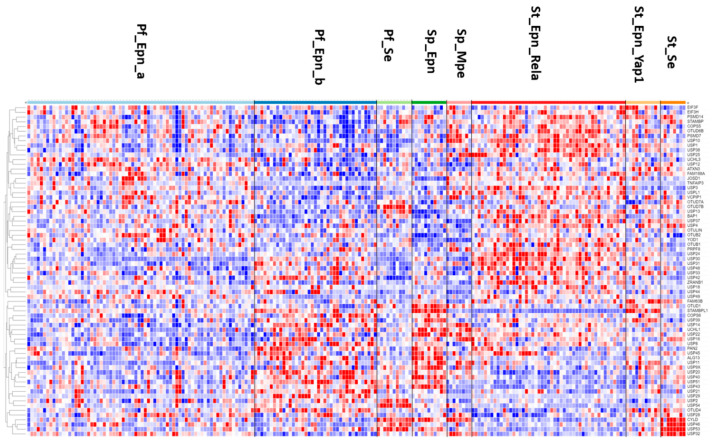
Heatmap and cluster analysis of DUB gene expression in ependymoma subtypes (Pf_Epn_a, N = 72; Pf_Epn-b, N = 39; Pf_se, N = 11; Sp_Epn, N = 11; Sp_Mpe, N = 8; St_Epn_Rela, N = 49; St_Epn_Yap1, N = 11; St_Se, N = 8).

**Figure 7 biomolecules-13-01503-f007:**
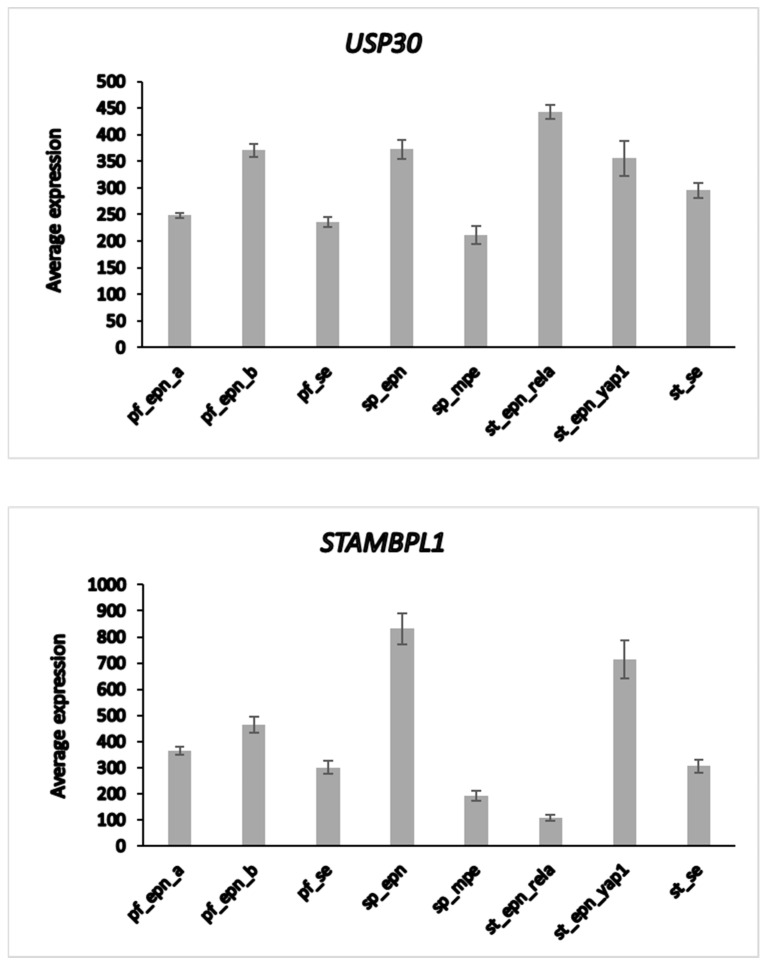
Differential expression of *USP30* and *STAMBPL1* in ependymoma molecular subgroups. *USP30*, F = 12.58, *p* = 2.38 × 10^−13^; *STAMBPL1*, F = 54.29, *p* = 5.23 × 10^−43^. This dataset is missing non-tumor control group for comparison by t-test.

**Figure 8 biomolecules-13-01503-f008:**
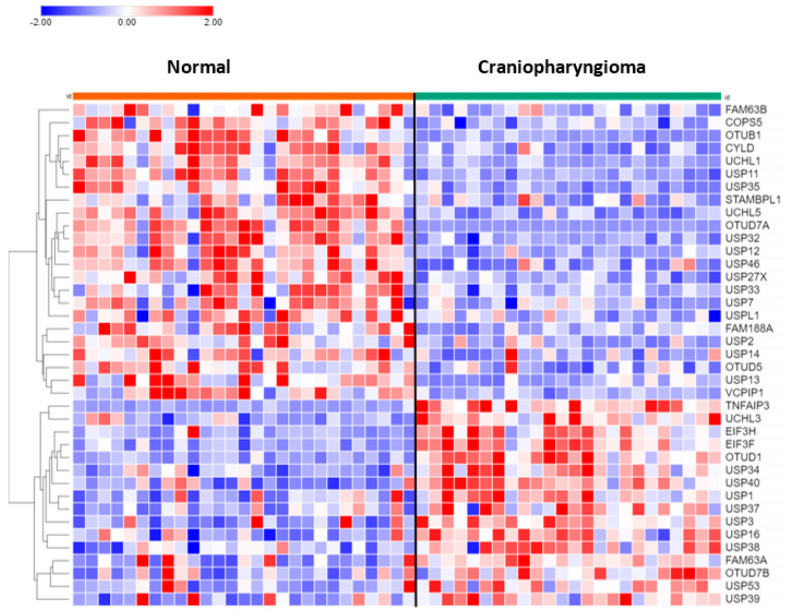
Heatmap and cluster analysis of DUB genes in craniopharyngioma. Normal, N = 27; Craniopharyngioma, N = 24.

**Figure 9 biomolecules-13-01503-f009:**
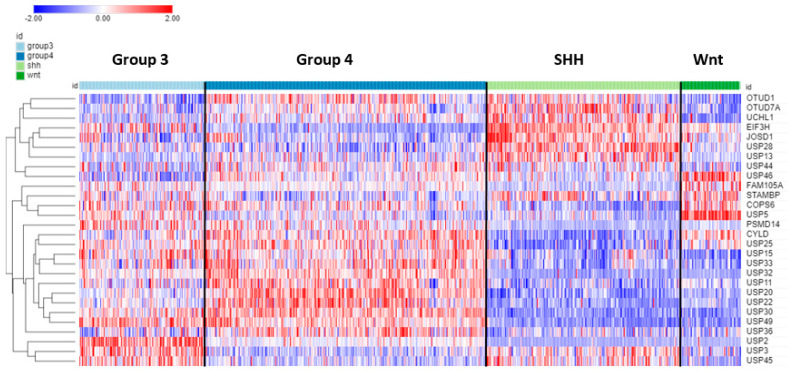
Heatmap and cluster analysis of the top 12 differentially expressed DUB genes for each of the 4 MB subgroups compared to non-tumor brain (note that due to overlap the total is less than 48). The Swartling dataset was used to identify top genes differing from non-tumor controls. The Cavalli dataset was used for the construction of the heatmap depicting the expression of these genes. Based on gene expression in the MB subgroups, the most significantly elevated DUBs for each group were Group 3—*USP2*; Group 4—*USP20*; SHH—*EIF3H*; WNT—*USP5.* Group 3, N = 144; Group 4, N = 326; SHH, N = 223; Wnt, N = 70).

**Figure 10 biomolecules-13-01503-f010:**
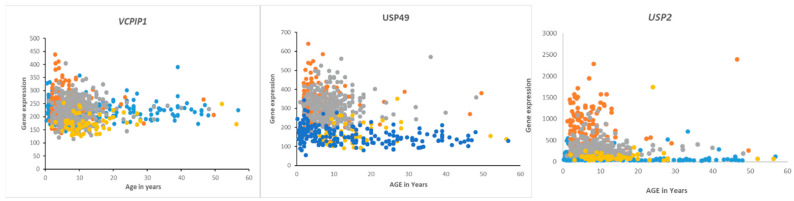
Gene expression of *VCPIP1*, USP49, and USP2 in medulloblastoma by age and subgroups. Orange—Group 3, Gray—Group 4, Blue—SHH, Yellow—Wnt. Of these genes, *USP2* expression was most specifically elevated in Group 3 MB infants and children compared to the other groups (Cavalli dataset) and compared to a non-tumor group (Swartling dataset).

**Figure 11 biomolecules-13-01503-f011:**
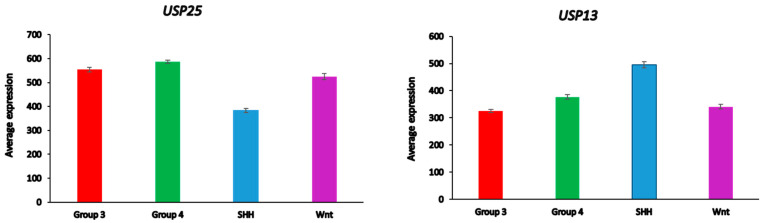
Differential expression of ERAD-associated DUBs in different medulloblastoma subgroups (Cavalli dataset). *USP25*, F = 138.98, *p* = 9.08 × 10^−72^; USP 13, F = 49.90, *p* = 1.93 × 10^−29^.

**Figure 12 biomolecules-13-01503-f012:**
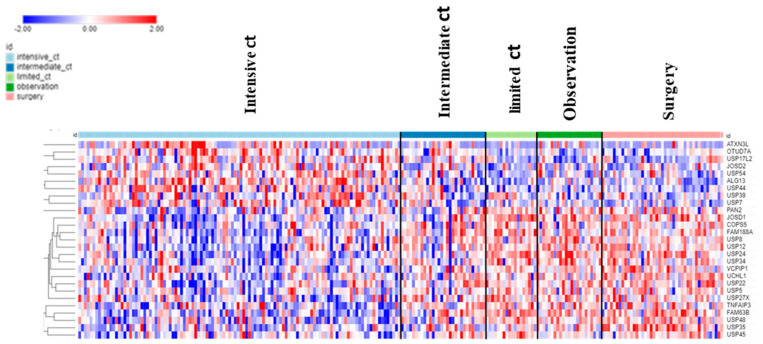
Treatment effects on DUB gene expression in Neuroblastoma. Intensive chemotherapy (ct) (N = 114), intermediate ct (N = 30), limited ct (N = 18), observation (no treatment control) (N = 23), surgery (N = 43).

**Figure 13 biomolecules-13-01503-f013:**
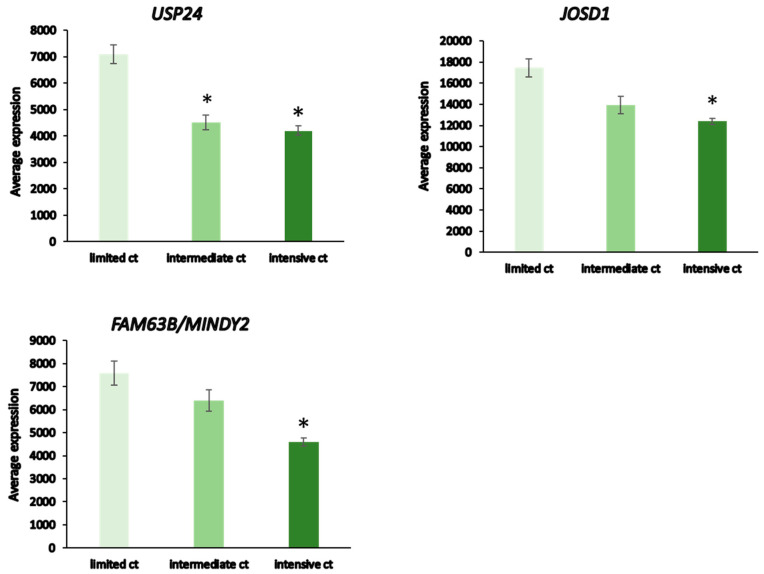
DUB expression in limited vs intensive chemotherapy of neuroblastoma from the Fischer dataset. Limited chemotherapy (ct), N = 18, intermediate ct, N = 30, intensive ct, N = 114. USP 24, F = 21.10, *p* = 7.44 × 10^−09^; JOSD1, F = 18.44, *p* = 6.26 × 10^−08^; MINDY2, F = 21.61, *p* = 4.99 × 10^−09^. * significantly different from limited ct group at *p* < 0.001 by *t*-test.

**Table 1 biomolecules-13-01503-t001:** Top ten differentially expressed DUB genes in gliomas by group compared to non-tumor group.

Astrocytoma ^1^	DUB	*p* Value (Corrected)	Chromosome	Versus Non-Tumor Group in the Sun Dataset
*CYLD*	Usp	2.44 × 10^−13^	16	Down
*USP12*	Usp	3.73 × 10^−12^	13	Down
*USP3*	Usp	1.95 × 10^−11^	15	Up
*OTUD7A*	Otu	1.26 × 10^−10^	15	Down
*OTUB1*	Otu	4.38 × 10^−10^	11	Down
*USP8*	Usp	1.69 × 10^−09^	15	Up
*USP33*	Usp	8.97 × 10^−09^	1	Down
*STAMBPL1*	Jamm	5.64 × 10^−08^	10	Down
*EIF3F*	Jamm	5.94 × 10^−08^	11	Up
*EIF3H*	Jamm	4.60 × 10^−07^	8	Up
**Glioblastoma ^2^**				
*USP12*	Usp	5.22 × 10^−20^	13	Down
*OTUD7A*	Otu	1.13 × 10^−19^	15	Down
*ZRANB1*	Otu	2.37 × 10^−18^	10	Down
*USP46*	Usp	3.60 × 10^−18^	4	Down
*OTUB1*	Otu	4.24 × 10^−18^	11	Down
*CYLD*	Usp	2.34 × 10^−16^	16	Down
*USP3*	Usp	1.20 × 10^−13^	15	Up
*USP27X*	Usp	5.35 × 10^−12^	X	Down
*USP30*	Usp	9.20 × 10^−12^	12	Down
*USP11*	Usp	1.39 × 10^−11^	X	Down
**Oligodendroglioma ^3^**				
*USP12*	Usp	3.27 × 10^−11^	13	Down
*OTUD7A*	Otu	3.55 × 10^−10^	15	Down
*USP48*	Usp	6.67 × 10^−09^	1	Down
*USP33*	Usp	1.37 × 10^−08^	1	Down
*CYLD*	Usp	1.45 × 10^−08^	16	Down
*USP3*	Usp	1.50 × 10^−08^	15	Up
*EIF3F*	Jamm	1.57 × 10^−08^	11	Up
*OTUD3*	Otu	2.46 × 10^−08^	1	Down
*USP14*	Usp	4.85 × 10^−08^	18	Down
*OTUB1*	Otu	6.68 × 10^−08^	11	Down

^1^ Total of 24 DUBs different from NT at *p* < 0.0001; ^2^ total of 43 DUBs different from NT at *p* < 0.0001; ^3^ total of 30 DUBs different from NT at *p* < 0.0001.

**Table 2 biomolecules-13-01503-t002:** DUB gene expression and survival data of glioma patients (French dataset).

DUB Gene	Family	Chromosome	Chi Square Kaplan Meier	*p* Value	Hazard Ratio (HR)	HR *p* Value	Better Survival
*ZRANB1*	Otu	10	116.69	3.36 × 10^−27^	0.21	4.6 × 10^−27^	High
*FAM188A*	Mindy	10	68.95	1.01 × 10^−16^	0.53	1.4 × 10^−13^	High
*USP34*	Usp	2	63.01	2.06 × 10^−15^	0.30	4.4 × 10^−12^	High
*USP49*	Usp	6	63.46	1.63 × 10^−15^	0.55	1.9 × 10^−16^	High
*Usp27X*	Usp	X	57.58	3.24 × 10^−14^	0.43	1.6 × 10^−15^	High
*USP54*	Usp	10	51.46	7.30 × 10^−13^	0.68	2.5 × 10^−11^	High
*USP51*	Usp	X	49.10	2.43 × 10^−12^	0.69	1.4 × 10^−11^	High
*USP30*	Usp	12	43.77	3.69 × 10^−11^	0.47	1.5 × 10^−08^	High
*USP11*	Usp	X	42.18	8.33 × 10^−11^	0.52	2.6 × 10^−10^	High
*OTUD7A*	Otu	15	38.15	6.54 × 10^−10^	0.69	1.3 × 10^−09^	High
*EIF3H*	Jamm	8	35.16	3.40 × 10^−09^	0.47	7.1 × 10^−10^	High
*USP1*	Usp	1	33.71	6.39 × 10^−09^	1.8	1.4 × 10^−08^	Low
*USP4*	Usp	3	33.07	8.90 × 10^−09^	2.6	5.6 × 10^−08^	Low
*ATXN3*	MJD	14	33.10	8.75 × 10^−09^	0.65	7.7 × 10^−06^	High
*USP43*	Usp	17	31.76	1.74 × 10^−08^	0.74	1.1 × 10^−08^	High
*USP46*	Usp	4	31.45	2.05 × 10^−08^	0.65	9.2 × 10^−08^	High
*TNFAIP3*	Otu	6	30.86	2.77 × 10^−08^	1.3	6.0 × 10^−07^	Low
*OTUD1*	Otu	10	28.27	1.06 × 10^−07^	0.54	2.0 × 10^−08^	High
*JOSD2*	MJD	19	25.88	3.62 × 10^−07^	1.5	2.1 × 10^−04^	Low
*PSMD7*	Jamm	16	24.22	8.60 × 10^−07^	1.9	1.3 × 10^−04^	Low
*STAMBPL1*	Jamm	10	23.87	1.03 × 10^−06^	0.76	4.3 × 10^−05^	High
*USP14*	Usp	18	23.95	9.90 × 10^−07^	2.1	1.0 × 10^−04^	low

**Table 3 biomolecules-13-01503-t003:** Differential expression of DUBs in ERAD signaling in glioma.

DUB Gene	Non-Tumor N = 23	Astrocytoma N = 26	Glioblastoma N = 77	Oligodendroglioma N = 50
*USP14*	851.87 ± 25.31	689.66 ± 25.63 *	738.5 ± 18.13 *	647.71 ± 16.28 *
*USP19*	159.11 ± 6.47	206.93 ± 9.5 *	222.11 ± 6.09 *	226.03 ± 7.9 *
*USP25*	314.57 ± 14.19	216.77 ± 9.34 *	216.88 ± 9.18 *	209.77 ± 7.59 *
*BAP1*	295.14 ± 10.78	221.5 ± 7.97 *	244.66 ± 5.02 *	248.25 ± 6.88 *

*USP14* (F = 12.27, *p* = 2.58 × 10^−07^), *USP19* (F = 10.53, *p* = 2.16 × 10^−06^), *USP25* (F = 14.59, *p* = 1.64 × 10^−08^), *BAP1* (F = 11.06, *p* = 1.12 × 10^−06^). * Significantly different from non-tumor group at *p* < 0.001.

**Table 4 biomolecules-13-01503-t004:** Top ten differentially expressed DUB genes by MB subgroup compared to non-tumor group in the Swartling dataset.

Group 3 (N = 233)	DUB	*p* Value vs. Non-Tumor	Chromosome	Versus Non-Tumor Group in the Swartling Dataset
*USP46*	Usp	1.16 × 10^−57^	4	Down
*USP2*	Usp	5.71 × 10^−53^	11	Up
*PSMD14*	Jamm	4.25 × 10^−51^	2	Up
*USP49*	Usp	1.34 × 10^−36^	6	Up
*USP28*	Usp	2.41 × 10^−26^	11	Down
*USP30*	Usp	1.29 × 10^−25^	12	Up
*UCHL1*	Uch	1.72 × 10^−22^	4	Down
*OTUD7A*	Otu	6.20 × 10^−22^	15	Down
*USP44*	Usp	6.39 × 10^−22^	12	Down
*COPS6*	Jamm	2.19 × 10^−19^	7	Up
**Group 4** **(N = 530**)				
*USP20*	Usp	8.96 × 10^−65^	9	Up
*USP28*	Usp	2.28 × 10^−63^	11	Down
*USP22*	Usp	1.35 × 10^−61^	17	Up
*USP32*	Usp	1.21 × 10^−58^	17	Up
*USP3*	Usp	2.98 × 10^−54^	15	Down
*USP30*	Usp	2.19 × 10^−52^	12	Up
*USP49*	Usp	1.49 × 10^−43^	6	Up
*USP45*	Usp	9.56 × 10^−36^	6	Down
*USP36*	Usp	7.95 × 10^−29^	17	Up
*EIF3H*	Jamm	1.59 × 10^−27^	8	Down
**SHH** **(N = 405)**				
*EIF3H*	Jamm	4.51 × 19^−135^	8	Up
*USP2*	Usp	1.17 × 10^−88^	11	Down
*USP20*	Usp	2.26 × 10^−69^	9	Down
*CYLD*	Usp	7.13 × 10^−64^	16	Down
*USP25*	Usp	4.63 × 10^−40^	21	Down
*USP32*	Usp	4.97 × 10^−38^	17	Down
*USP33*	Usp	5.42 × 10^−36^	1	Down
*JOSD1*	Mjd	2.40 × 10^−30^	22	Up
*USP11*	Usp	3.32 × 10^−28^	X	Down
*USP13*	Usp	1.03 × 10^−26^	3	Up
**WNT** **(N = 118)**				
*UCHL1*	Uch	2.14 × 10^−82^	4	Down
*USP2*	Usp	3.18 × 10^−56^	11	Down
*USP20*	Usp	6.88 × 10^−50^	9	Down
*USP32*	Usp	2.23 × 10^−47^	17	Down
*USP5*	Usp	1.19 × 10^−38^	12	Up
*COPS6*	Jamm	2.59 × 10^−32^	7	Up
*EIF3H*	Jamm	1.19 × 10^−29^	8	Up
*OTUD7A*	Otu	1.48 × 10^−29^	15	Down
*USP33*	Usp	1.38 × 10^−28^	1	Down
*USP28*	Usp	6.05 × 10^−27^	11	Down

**Table 5 biomolecules-13-01503-t005:** DUB gene expression and medulloblastoma survival data (Cavalli dataset).

DUB	DUB Family	Chromosome	Chi Square Kaplan Meier	*p* Value	Hazard Ratio	*p* Value for Hazard Ratio	Better Survival
*VCPIP1*	Otu	8	28.14	1.13 × 10^−07^	1.9	1.10 × 10^−05^	Low
*USP49*	Usp	6	26.73	2.34 × 10^−07^	1.9	1.30 × 10^−05^	Low
*USP2*	Usp	11	21.82	2.99 × 10^−06^	1.3	3.50 × 10^−05^	Low
*USP51*	Usp	X	19.36	1.08 × 10^−05^	0.44	1.00 × 10^−04^	High
*STAMBPL1*	Jamm	10	16.70	4.37 × 10^−05^	0.73	3.10 × 10^−04^	High
*PSMD14*	Jamm	2	18.05	2.16 × 19^−05^	1.4	4.20 × 10^−04^	Low
*ZRANB1*	Otu	10	17.11	3.52 × 10^−05^	0.54	1.00 × 10^−03^	High
*OTUD3*	Otu	1	18.76	1.48 × 10^−05^	2.4	1.30 × 10^−03^	Low
*PRPF8*	Jamm	17	18.77	1.47 × 10^−05^	0.58	2.20 × 10^−03^	High
*USP15*	Usp	12	16.42	5.06 × 10^−05^	2	3.00 × 10^−03^	Low
*USP45*	Usp	6	14.82	1.18 × 10^−04^	1.5	3.90 × 10^−03^	Low
*USP26*	Usp	X	22.77	1.82 × 10^−06^	7.3	5.20 × 10^−03^	Low
*USP36*	Usp	17	22.07	2.63 × 10^−06^	2.2	5.50 × 10^−03^	Low
*USPL1*	Usp	13	14.89	1.14 × 10^−04^	2.1	5.80 × 10^−03^	Low
*USP25*	Usp	21	12.36	4.39 × 10^−04^	1.6	8.70 × 10^−03^	Low
*EIF3H*	Jamm	8	13.23	2.75 × 10^−04^	1.5	9.40 × 10^−03^	Low
*COPS5*	Jamm	8	9.25	2.36 × 10^−04^	1.9	9.90 × 10^−03^	Low

**Table 6 biomolecules-13-01503-t006:** DUBs and GO category of “Regulation of immune response” by MB group compared to non-tumor group of Swartling dataset (n = 291).

DUB	Family	*p* Value Corrected FDR	Versus Non-Tumor Group in Swartling Dataset
**Group 3 (n = 233)**			
*PSMD14*	Jamm	8.74 × 10^−52^	Up
*OTUD7A*	Otu	1.70 × 10^−22^	Down
*TNFAIP3*	Otu	6.26 × 10^−10^	Up
*CYLD*	Usp	3.54 × 10^−08^	Down
**Group 4 (n = 530)**			
*PSMD14*	Jamm	2.23 × 10^−26^	Up
*OTUD7A*	Otu	1.16 × 10^−13^	Down
*CYLD*	Usp	7.04 × 10^−10^	Up
*TNFAIP3*	Otu	5.67 × 10^−09^	Up
**SHH (n = 405)**			
*CYLD*	Usp	1.95 × 10^−64^	Down
*PSMD14*	Jamm	4.95 × 10^−09^	Down
*TNFAIP3*	Otu	2.95 × 10^−08^	Up
*PSMD7*	Jamm	5.19 × 10^−08^	Up
**Wnt (n = 118)**			
*OTUD7A*	Otu	8.10 × 10^−30^	Down
*PSMD7*	Jamm	3.89 × 10^−25^	Up
*TNFAIP3*	Otu	2.40 × 10^−10^	Down
*CYLD*	Usp	4.19 × 10^−04^	Up
*PSMD14*	Jamm	8.99 × 10^−04^	Up

**Table 7 biomolecules-13-01503-t007:** DUBS and GO category of DNA repair in MB groups compared to non-tumor group.

DUB	Family	*p* Value vs. NT Group Swartling	Versus Non-Tumor Group in Swartling Dataset
**Group 3 vs. NT**			
*PSMD14*	Jamm	2.27 × 10^−51^	Up
*USP28*	Usp	1.07 × 10^−26^	Down
*COPS6*	Jamm	1.30 × 10^−21^	Up
*USP47*	Usp	3.96 × 10^−15^	Down
*UCHL5*	Uch	6.12 × 10^−14^	Up
*COPS5*	Jamm	1.61 × 10^−10^	Up
*USP1*	Usp	2.53 × 10^−06^	Up
*OTUB1*	Otu	3.33 × 10^−05^	Down
**Group 4 vs. NT**			
*Usp28*	Usp	8.12 × 10^−64^	Down
*USP3*	Usp	1.33 × 10^−54^	Down
*USP45*	Usp	4.54 × 10^−36^	Down
*PSMD14*	Jamm	1.45 × 10^−26^	Up
*COPS6*	Jamm	3.10 × 10^−24^	Up
*USP1*	Usp	1.48 × 10^−16^	Up
*OTUB1*	Otu	1.67 × 10^−14^	Down
*USP7*	Usp	2.65 × 10^−12^	Up
*COPS5*	Jamm	4.01 × 10^−06^	Down
*USP47*	Usp	2.16 × 10^−04^	Down
*UCHL5*	Uch	2.11 × 10^−03^	Up
**SHH vs. NT**			
*USP10*	Usp	2.21 × 10^−15^	Up
*PSMD14*	Jamm	8.58 × 10^−09^	Down
*COPS5*	Jamm	1.16 × 10^−08^	Up
*USP45*	Usp	3.60 × 10^−07^	Down
*UCHL5*	Uch	1.30 × 10^−06^	Down
*USP3*	Usp	5.90 × 10^−06^	Down
*COPS6*	Jamm	5.98 × 10^−06^	Down
*USP1*	Usp	6.06 × 10^−06^	Up
*USP47*	Usp	6.68 × 10^−04^	Down
*USP7*	Usp	1.71 × 10^−03^	Up
**WNT vs. NT**			
*COPS6*	Jamm	2.67 × 10^−32^	Up
*USP28*	Usp	5.39 × 10^−27^	Down
*USP3*	Usp	3.53 × 10^−25^	Down
*USP45*	Usp	2.61 × 10^−22^	Down
*UCHL5*	Usp	4.72 × 10^−14^	Down
*USP10*	Usp	7.73 × 10^−11^	Up
*COPS5*	Jamm	1.02 × 10^−03^	Up
*PSMD14*	Jamm	1.46 × 10^−03^	Up
*OTUB1*	Otu	2.89 × 10^−03^	Up
*USP1*	Usp	4.55 × 10^−03^	Up

## Data Availability

The data referred to in this manuscript are publicly available at the R2 Genomics Analysis and Visualization Platform (http://r2.amc.nl (accessed on 5 September 2023)) and at the NIH GEO database and are available upon reasonable request from the first author.

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
