# Peer review of "DUBing Primary Tumors of the Central Nervous System: Regulatory Roles of Deubiquitinases"

_biomolecules, 2023, doi:10.3390/biom13101503_

Round 1
Reviewer 1 Report
The authors present a complete, interesting, and well-written review article describing the regulatory role of deubiquitinases in brain tumorigenesis. Despite there being several studies on this topic, the authors manage to separate themselves from the rest of the articles with an in silico approach that shows the potential of several deubiquitinases in the regulation of important cellular functions (UPR, ERAD, immune response, DNA repair), discussing putative applications as biomarkers or therapeutic targets. The findings are novel and invite the development of multiple downstream studies for the in vitro and in vivo verification of these data. I do not consider adding or removing content necessary and only leave two minor revisions. Congratulations.
-Indicate in the Figure 2 legend that the symbol ‘*’ shows the difference between GBM and AS.
-The authors should highlight that some deubiquitinases not described/mentioned in this manuscript have an already described function in regulating brain tumorigenesis. These recent two reviews mention and describe these DUBs and should be referenced in this study (doi: 10.14348/molcells.2020.2289; DOI: 10.1007/s12035-021-02339-4).
Reviewer 2 Report
In the manuscript ''DUBing primary tumors of the central nervous system: Regulatory roles of deubiquitinases'' the authors utilize various data bases to investigate the expression patterns of the various members of the DUB family in several malignancies of the CNS.DUBs are a family of the ubiquitin editing enzymes of approximately 99 members, divided into 5 main major groups depending on sequence similarities in the active site. DUBs function to regulation the stability, activity or location of many proteins involved in cancer progression. Although over or under expression of DUBs has been observed in several tumour types, the relationship between DUB expression and CNS tumours is relatively unexplored. Through a series of analysis, the authors identify dubs showing differential expression in a panel of gene expression from several CNS tumours. The authors do an impressive job of analysing the available data sets and presenting the most relevant findings across multiple CNS types. As such this article maybe of interest to researchers in the CNS and ubiquitin fields however I have some reservations and comments.
Considering the amount of data analyzed I am not sure this qualifies as a review article as new information is presented by the authors. In my opinion it could be broken down into several bioinformatic studies on specific CNS diseases. As it stands it is difficult to interpret the significance of the findings across the multiple tumor types.
I think the introduction, while comprehensive, is a little too focused on aspects of organelle function and role of specific stress pathways. Considering the review is focused on CNS I feel that more space should be given over to the different types of CNS tumors and what is known (if anything about the UPS in these tumors).
The section on chromosome 10 feels a bit premature since there is no data linking loss of chromosome 10 in the samples analyzed. Although it could be suggested as part of a discussion it feels a bit out of place in the results section.
Small comments:
Line 39: I think UBA6 is also classified as an E1 ligase bringing the number to two.
Line 40: I think it would be clearer to refer to the E3 ligases based on the RING or HECT domain families instead or ligase activity and adaptors,
The quality of language is fine
Reviewer 3 Report
The reviewed manuscript entitled: “DUBing primary tumors of the central nervous system: Regulatory roles of deubiquitinases” presents a comprehensive bioinformatic analysis of deubiquitinases in the central nervous system tumors. The role of DUBs in brain tumors in not fully understood, and requires further investigation. Taking into consideration high mortality of brain tumors and the need for novel diagnostic and treatment options, the analysis that was done by the Authors seems of high importance.
In this article the Authors performed a bioinformatic analysis determining the expression of DUB genes in publicly available datasets of brain tumors. They also plotted gene expression by tumor subgroup and by age for each DUB which enabled the comparison of gene expression in pediatric vs adult age groups. They also established the relationship of DUBs with ERAD, UPR, DNA damage repair pathways, and discussed their suitability as potential biomarkers and therapeutic targets.
The article is well structured and nicely written. It provides a lot of valuable data potentially relevant to the diagnosis and treatment of brain tumors. Thus, I recommend it for publication, once minor revision is done by the Authors.
My only concern in that the results of the bioinformatic analysis should always be confirmed/validated by biological experiments. However I assume it can be done by other research groups.
Minor comments:
Line 264 – typo or missing p value
Line 289– “OTUD7A expression was significantly depressed in AS, GBM, and ODG” – maybe instead of “depressed” use the word “decreased” or “diminished”? Same in line 333, 378 etc.
Line 294 – remove italics from the word “and”
Lines 301, 361, 395, 472 - NF-kB – should be NF-ĸB
Figures should be replaced with better quality ones (the font is not the easiest to read and the bright colors are a bit disturbing). I suggest using e.g. GraphPad Prism instead of Excel – it will look way better.
Line 310 – non-tumor or non tumor (in Figure 4) – use the hyphen consistently; Line 325 - Non tumor – why capital letter?
Line 316, 389 etc. - OTUD7A or OTUD7a (line 299) – use the correct form consistently
Line 341 - reported to be expressed higher in cancer tissue à reported to have higher expression
Line 365 - Among the significant differentially expressed DUBs in the EPN dataset were five genes associated with the Regulation of ERAD pathway, including ATXN3, USP3, USP14, USP25, and OTUD2/YOD1 à there were
Line 492 – DUBS à DUBs
Line 658 – relevant used twice in one sentence
Generally high quality of English. Check for typos etc.
Round 2
Reviewer 2 Report
I appreciate the authors response and and happy to recommend publication.